# Chunk Based Speech Pre-training with High Resolution Finite Scalar Quantization

## Abstract

Low latency speech human-machine communication is becoming increasingly necessary as speech technology advances quickly in the last decade. One of the primary factors behind the advancement of speech technology is self-supervised learning. Most self-supervised learning algorithms are designed with full utterance assumption and compromises have to made if partial utterances are presented, which are common in the streaming applications. In this work, we propose a chunk based self-supervised learning (Chunk SSL) algorithm as an unified solution for both streaming and offline speech pre-training. Chunk SSL is optimized with the masked prediction loss and an acoustic encoder is encouraged to restore indices of those masked speech frames with help from unmasked frames in the same chunk and preceding chunks. A copy and append data augmentation approach is proposed to conduct efficient chunk based pre-training. Chunk SSL utilizes a finite scalar quantization (FSQ) module to discretize input speech features and our study shows a high resolution FSQ codebook, i.e., a codebook with vocabulary size up to a few millions, is beneficial to transfer knowledge from the pre-training task to the downstream tasks. A group masked prediction loss is employed during pre-training to alleviate the high memory and computation cost introduced by the large codebook. The proposed approach is examined in two speech to text tasks, i.e., speech recognition and speech translation. Experimental results on the LIBRISPEECH and MUST-C datasets show that the proposed method could achieve very competitive results for speech to text tasks at both streaming and offline modes.

## 1 Introduction

Transcribe speech into text in real time is critical for applications requiring immediate feedback, such as real-time transcription of broadcast contents, voice assistants, and simultaneous speech translation etc. It requires processing audio incrementally as it is received, rather than waiting for the entire utterance to be available. For neural network based end-to-end systems, the requirement includes two-fold. First, a speech encoder, such as causal encoder (Zhang et al., 2020b) and chunk encoder (Tsunoo et al., 2019; Shi et al., 2021), is able to process input audio cumulatively without dependency on the future input. Second, a decoder is required to decode transcription incrementally based on partial encoder outputs. Frame-Synchronous decoders (Dong et al., 2020), such as CTC (Graves et al., 2013) Transducer (Graves, 2012) CIF (Dong & Xu, 2020) and TAED (Tang et al., 2023), are capable of generating transcription based on partial encoder outputs and meet the streaming requirement naturally.

Self-supervised pre-training leverages vast amounts of unlabeled data and learn universal feature representations for downstream tasks, especially for tasks with limited supervised training data. Speech pre-training methods have emerged as the backbone of many speech processing tasks (Chung et al., 2018; Baevski et al., 2020b; Hsu et al., 2021b; Chen et al., 2021; Chiu et al., 2022; Tang et al., 2022; Baade et al., 2025). Those methods are designed for the speech applications with full utterance available, and compromises have to be made if the downstream tasks are streaming applications. Chiu et al. (2022) propose to conduct pre-training with a causal encoder, which sacrifices right context information. Fu et al. (2024) modify the encoder structure and conduct continual pre-training to adapt the model for streaming applications.

In all those methods aforementioned, a dedicated pre-trained model has to be built for the streaming scenario instead of sharing the same model with the offline application.

In this study, we focus on building an encoder suitable for both streaming and offline modes, with a chunkwise self-supervised learning (Chunk SSL) framework as depicted in Figure 1. Chunk SSL aims to restore discrete indices of the masked frames based on unmasked frames in the last chunk and previous chunks. The discrete indices are estimated with finite scalar quantization (Mentzer et al., 2024) (FSQ) with vocabulary size up to millions (§3).The pre-training FSQ token has more fine-grained resolution compared with the downstream modeling units, such as phonemes or sentencepiece

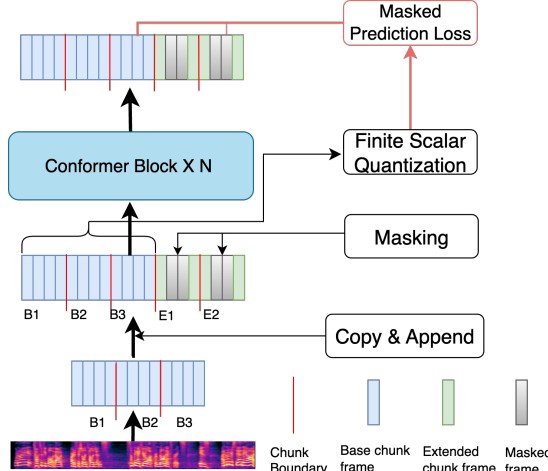

Figure 1: Chunkwise self-supervised training.

tokens, which usually are with vocabulary size ranging from tens to tens of thousands. We hypothesize that speech frames associated with a high resolution FSQ token are mainly mapped to an unique modeling unit in the downstream task and it makes the knowledge transfer easier from the pre-training stage to the fine-tuning stage. However, those high resolution FSQ tokens pose a great challenge for modeling and we propose to decompose a large codebook into small channel based sub-codebooks to alleviate the memory and computation cost during pre-training. Details could be found in §3.

In Figure 1, speech features are first extracted from the input audio, and grouped into equal sized chunks. Instead of calculating those chunks from left to right sequentially, a copy and append data augmentation (CADA) is introduced to parallelize the computation for all chunks in the same utterance (§2). CADA augments the input sequence by copying input chunks and appending them to the end of utterance as extended chunks. Masking is applied to frames in extended chunks only. Augmented utterances are then processed by a CADA compatible Conformer encoder, which could handle those augmented chunks properly even though frames in extended chunks are with altered location information. Finally, the model is optimized by restoring FSQ indices of those masked frames from Conformer encoder outputs. Experiments on LIBRISPEECH and MUST-C datasets indicate that the same model initialized with Chunk SSL could achieve very competitive results on both streaming and offline speech to text tasks. It shows that Chunk SSL eliminates the need to build a dedicated streaming and dedicated offline model. To summarize, our contributions includes:

1. We propose a chunkwise speech self-supervised learning algorithm for both streaming and offline speech to text tasks.

2. A copy and append data augmentation improves the pre-training efficiency by reusing the chunk level computation results and parallelizing the chunkwise based computation.

3. FSQ is employed to generate high resolution speech codebook and a group masked prediction loss is proposed to alleviate the computation challenge.

4. Our results show the proposed method can build one model for both scenarios and achieve competitive results on the LIBRISPEECH and MUST-C datasets.

## 2 COPY AND APPEND DATA AUGMENTATION

The naive Chunk SSL algorithm, which recovers masked frame indices in the right most chunk based on the unmasked frames in the same chunk and preceding chunks, is executed chunk by chunk in a sequential order instead of computing all chunks from one utterance in a parallel fashion. Inspired by the implementation of the chunk encoder with a look-ahead chunk for the streaming speech translation (Liu et al., 2021), we propose a copy and append data augmentation for the Chunk SSL,

which reorganizes input data and changes the augmented data computation, but it is still strictly equivalent to the computation in the naive Chunk SSL algorithm.

The pre-training procedure is described in Figure 1. Input features are first downsampled with a stacked subsampling module. The outputs are then divided into chunks with fixed duration, They are considered as base chunks. The consecutive chunk of each base chunk is copied and appended to the end of the utterance in a sequential order. Those newly created chunks are called extended chunks. For example, there are 12 speech input frames in Figure 1 after subsampling. Assuming the chunk size is 4 frames, the speech features are segmented into three base chunks (in cyan): "B1", "B2" and "B3". Chunk "B2" and "B3" are copied and appended to the end of the utterance as "E1" and "E2" (in lime). They correspond to the extended chunks of "B1" and "B2" respectively. There is no extended chunk for the last chunk ("B3"). Extended chunks act as right most chunks with different preceding chunks in the naive Chunk SSL algorithm and masking is only applied on frames of extended chunks.

A speech encoder, such as Transformer (Vaswani et al., 2017) and Conformer (Gulati et al., 2020), contains two types of modules. One is the intra-frame module where the computation is based on one frame and no location information or other frames are required, for example, LayerNorm and feedforward layers. The other type is the inter-frame module where the computation is related to the input location and requires information from neighbouring frames. The inter-frame modules include self-attention and convolutional layers. An augmented utterance aforementioned, can be processed as a normal utterance in the intra-frame modules and no modification is required. On the other hand, the position information is invalid among extended chunk frames, since they are copied and appended to the end of utterance. Modifications are introduced for two inter-frame modules: self-attention and convolutional layers, to handle inconsistent position information in augmented utterances as described in the following subsections.

## 2.1 CADA SELF-ATTENTION MODULE

The self-attention module exchanges information within the same sequence according to the attention weights. Assume $\boldsymbol{X} = \{\boldsymbol{x}_0, \boldsymbol{x}_1, \cdots, \boldsymbol{x}_{N-1}\}$ is a $N$ length input sequence to a self-attention layer, the attention weights $\boldsymbol{A} = \{A_{i,j}\}$ are calculated as

$$A_{i,j} = \frac{\exp\big(\beta(\boldsymbol{W}_q\boldsymbol{x}_i)^T(\boldsymbol{W}_k\boldsymbol{x}_j) - \zeta(1 - \alpha_{i,j})\big)}{\sum_{\tau} \exp\big(\beta(\boldsymbol{W}_q\boldsymbol{x}_i)^T(\boldsymbol{W}_k\boldsymbol{x}_\tau) - \zeta(1 - \alpha_{i,\tau})\big)} \tag{1}$$

where $\beta = \frac{1}{\sqrt{d}}$ is a scaling factor, $\zeta$ is a big number (1e6), $\boldsymbol{W}_q, \boldsymbol{W}_k \in \mathcal{R}^{d \times d}$ are transform matrices, and $d$ is the dimension of model embedding[1]. $\boldsymbol{\alpha} = \{\alpha_{i,j}\}, i, j \in [0, N-1]$ is a masking matrix with value 1 or 0 for each component and $\alpha_{i,\tau} = 0$ means $\boldsymbol{x}_\tau$ will not contribute the computation of attention weights $A_{i,*}$ and the corresponding attention output of $\boldsymbol{x}_i$. The self-attention module leverages the attention mask $\alpha$ to control information exchange within frames from the same sequence. For example, an encoder usually sets all $\alpha_{i,\tau} = 1$ for full information access.

Assuming $N$ is the number of input frames in an utterance and is a multiple of chunk size $C$. The augmented input sequence length is $N' = 2N - C$, the number of base chunks is $M = N/C$ and the number of extended chunks is $M - 1$. We define $m(i) = \lfloor i/C \rfloor$ as the chunk index of frame $i$. If the left context is infinite, i.e., the model could access all history information, a CADA masking matrix $\boldsymbol{\alpha}$ is defined as

$$\alpha_{i,j} = \begin{cases} 1, & \text{if } m(i) > m(j) \text{ and } i < N \\ & \text{or if } m(i) == m(j) - M \\ & \text{or if } m(i) == m(j) \\ & \text{or if } m(i) \geq m(j) + M \\ 0, & \text{otherwise} \end{cases} \tag{2}$$

where $i, j \in [0, N' - 1]$. The first two rows in Eq. (2) define the masking matrix for a frame $i$ in base chunks and the fourth row is for frames in extended chunks only. To be more specific,

1. the first row means a frame $i$ from base chunks could access all information in the previous chunks

---

[1]Single head attention is discussed for simplicity.

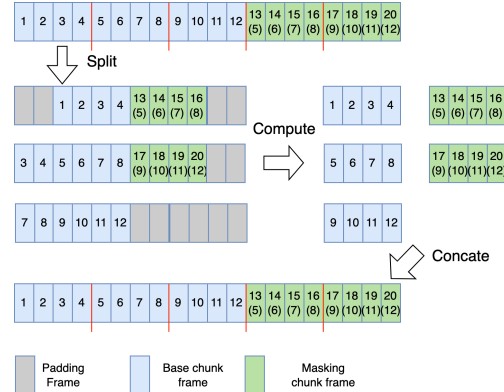

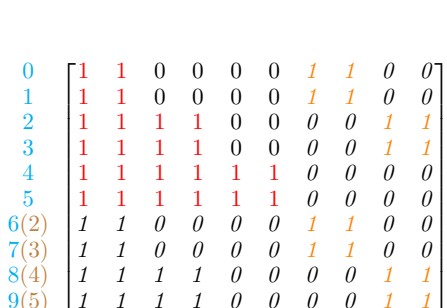

(a) Self-Attention CADA Masking for the augmented utterance with original sequence length 6 and chunk size 2.

(b) Pair, compute and concatenate convolutional computation for the CADA sequence with chunk size 4 and $L_{lc} = L_{rc} = 2$.

Figure 2: Illustration of CADA sequence computation.

2. the second row represents a frame $i$ from base chunks can also access limited future information from its extended chunk

3. the third row indicates a frame $i$ could access all frames within the same chunk

4. the fourth row means a frame $i$ from extended chunks can access all information in its base chunk and preceding chunks of the base chunk

5. there is no information exchange for all other cases as shown in the fifth row.

Figure 2a depicts a CADA masking example for an utterance with 6 frames before CADA. The first column is the frame index. The number in the bracket (brown) stands for the extended chunk frame's original position, i.e., the position of the base chunk frame that the frame is copied from. $\alpha$ for the extended chunks are presented in italic fonts. A self-attention module equipped with a CADA masking matrix can process a CADA sequence with position information preserved.

## 2.2 CADA CONVOLUTIONAL MODULE

A convolutional layer has a set of filters which slide across input features within the same utterance and extract localized representations. A convolutional layer assumes the input frames organized as consecutively, which no longer holds true for extended chunk frames in a CADA utterance. This problem can be resolved via a chunk based convolutional computation (Li et al., 2023). Without loss of generality, we assume the convolution module is a depth-wise convolution layer with $L_{lc}$ left and $L_{rc}$ right context frames. The modified computation includes three steps. First, we separate the CADA utterance into chunks and pair a base chunk and its extended chunk together, for example "B1" and "E1" in Figure 1; then the last $L_{lc}$ frames from the chunk preceding the base chunk are appended to the left as left context. If the base chunk is the first chunk, $L_{lc}$ zero padding frames are attached. Similarly, $L_{rc}$ zero padding frames are appended to the right and it leads to a new concatenated subsequence with length $L_{lc} + 2C + L_{rc}$. Second, the concatenated subsequence is fed to the convolution module and obtains output subsequence with length $2C$. $L_{lc}$ and $L_{rc}$ context frames are absorbed during convolution computation. The first $C$ output frames correspond to the base chunk and the second $C$ frames are for the extended chunk. Finally, those output chunks are placed back to their original positions in the augmented utterance as the outputs from the CADA convolutional module. Figure 2b demonstrates three steps to conduct convolution computation on an augmented utterance with base sequence length 12 and chunk size 4.

## 2.3 DYNAMIC CHUNK BASED PRE-TRAINING

Dynamic chunk training (Zhang et al., 2020a; Weninger et al., 2022; Li et al., 2023) is a useful fine-tuning approach for the chunk based ASR encoder. During training, the chunk duration varied

from hundreds of million seconds to a few seconds is randomly chosen for every model update. A model built with dynamic chunk training is suitable for both streaming and offline modes. We extend dynamic chunk training for the Chunk SSL pre-training. Chunk duration is chosen from 6 durations ($[640, 1280, 1920, 2560, 3200, 3840]$ million seconds) randomly for every model update.

# 3 HIGH RESOLUTION FINITE SCALAR QUANTIZATION

## 3.1 QUANTIZATION MODULE

As shown in Figure 1, we leverage finite scalar quantization (Mentzer et al., 2024) to quantize input frames and obtain their discrete indices. A FSQ module is built prior to the Chunk SSL pre-training. In the quantization module, an utterance based channel mean and variance normalization is first applied to input frames $\boldsymbol{X}$ to obtain $\hat{\boldsymbol{X}} = \{\hat{\boldsymbol{x}}_i\}$, then it is processed by a FSQ encoder $\text{Enc}_f$ and projected into a low-dimension space with size $d'$, where $d' \ll d$. The output of each channel $r \in [1, d']$ for input frame $i$ is rounded to an integer $h_{i,r}$

$$h_{i,r} = \text{ROUND}\big(\lfloor K_r/2 \rfloor \tanh(\text{Enc}_f(\hat{\boldsymbol{x}}_i)[r]\big), \text{ and } h_{i,r} \in [-\lfloor K_r/2 \rfloor, \lfloor K_r/2 \rfloor] \tag{3}$$

where ROUND is a bounded round operation and $K_r$ is the corresponding number of levels at channel $r$. Then the output $\boldsymbol{h}_i = \{h_{i,r}\}$ is projected back to the original space via a decoder $\tilde{\boldsymbol{x}}_i = \text{DEC}_f(\boldsymbol{h}_i)$. The FSQ module is optimized by matching reconstructed $\tilde{\boldsymbol{x}}_i$ and input $\hat{\boldsymbol{x}}_i$.

## 3.2 GROUP MASKED PREDICTION LOSS

The acoustic encoder is optimized with masked prediction loss. We encourage the Chunk SSL model to reconstruct the masked frames based on the context information provided. More specifically, we mask half consecutive frames in every extended chunk. The start position of the masking frame is randomly chosen from $[0, \frac{C}{4}]$ at every extended chunk, so we have enough unmasked frames on both sides of the masked frames as context to infer those masked frames.

Assuming the Chunk SSL model outputs for the augmented sequence is $\boldsymbol{O} = \{\boldsymbol{o}_i\}$. The FSQ index of $x_i$ is denoted as $\mu_i$ and $\mathbf{e}_{\mu_i}$ is the output embedding of $\mu_i$. The masked prediction loss is defined as

$$\mathcal{L}_m = -\sum_{i \in \boldsymbol{M}} \log \frac{\exp(\boldsymbol{o}_i^T \boldsymbol{e}_{\mu_i})}{\sum_{j=1}^{V} \exp(\boldsymbol{o}_i^T \boldsymbol{e}_j)} \tag{4}$$

where $\boldsymbol{e}_j$ is the $j$-th output embedding, $\boldsymbol{M}$ is the masked frames set, and $V$ is vocabulary size from the quantization module.

We hypothesize a high resolution FSQ codebook would be beneficial for downstreaming tasks due to two reasons. First, FSQ does not suffered from codebook collapse and can achieve high codebook usage. Second, a high resolution codebook divides the feature space into more fine-grained bins and frames sharing the same token index might be more closer to each other compared with frames associated with a token index from a low resolution codebook. When the FSQ codebook is large enough, each FSQ token could be mainly associated with one modeling unit, such as phoneme, in the downstreaming task, hence it will make the knowledge transfer easier from the pre-training stage to the fine-tuning stage. This hypothesis is examined in §5.2. However, a high resolution FSQ codebook poses a great challenge for optimization. For example, the codebook with vocabulary size 1,265,625 and embedding size 512 would take about 2.4G memory if they are stored in float data type. In order to alleviate this issue, we propose to decompose the codebook into a group of channel based sub-codebooks and compute prediction loss for each group one by one. Given frame $i$, the sub-codebook index at $r$th channel is $h_{i,r}$. We define the group masked prediction loss as

$$\mathcal{L}_m = -\sum_{i \in \boldsymbol{M}} \sum_{r} \log \frac{\exp(\boldsymbol{o}_i^T \boldsymbol{e}_{h_{i,r}}^r)}{\sum_{j=1}^{K_r} \exp(\boldsymbol{o}_i^T \boldsymbol{e}_j^r)} \tag{5}$$

where $\boldsymbol{e}_j^r$ is the $j$th output embedding at the $r$th channel sub-codebook. Optimization on sub-codebook individually is equivalent to optimizing the full codebook, since a perfect system that accomplishes the sub-tasks in Eq. (5) could solve the Eq. (4) too, but with much less memory requirement. For the same codebook with number of levels per channel $[5, 5, 5, 5, 5, 5, 3, 3, 3, 3]$ and size 1,265,625 aforementioned, they only take about 84k storage memory.

# 4 EXPERIMENTAL SETTINGS

## 4.1 DATA

**Pre-training**: There are two pre-training tasks in this work, i.e., FSQ SSL training and Chunk SSL training. We first build FSQ module to discretize the input speech feature and then pre-train speech encoder with Chunk SSL to generate contextual representation. LIBRI-LIGHT (Kahn et al., 2019), which includes 60k hours of unlabelled English speech data, is used in both pre-training tasks.

**Fine-Tuning**: For speech recognition task, the pre-trained models are finetuned and evaluated on LIBRISPEECH (Panayotov et al., 2015) dataset. We use two dev sets for development and report all results from dev and test sets. For speech translation task, experiments are conducted on two MUST-C (Gangi et al., 2019) language pairs: English to German (EN→DE) and English to Spanish (EN→ES). Sequence level knowledge distillation (Kim & Rush, 2016) is applied to boost the speech translation quality. The models are developed on the dev set, and the final results are reported on the tst-COMMON set.

## 4.2 MODEL CONFIGURES

The speech encoder is a Conformer (Gulati et al., 2020) based chunk encoder. The encoder is equipped with a relative positional embedding (Shaw et al., 2018) and starts with a stacking layer to down-sample the input features by four times. Two model configurations: base and large, have been explored. The base encoder has 12 Conformer layers, input embedding size of 512, 8 attention heads, feedforward layer dimension 2048 and convolutional module kernel size 31. The large encoder has 24 Conformer layers, input embedding size of 768, 16 attention heads, feedforward layer dimension 3072 and convolutional module kernel size 5. Transducer is adopted in the fine-tuning experiments. The predictor module is with one Transformer layer (Vaswani et al., 2017) for speech recognition tasks and two layers for speech translation tasks. The input embedding and feedforward layer dimension are the same as ones in the encoder setting if not mentioned specifically. The joint module is a feedforward layer as (Zhang et al., 2020b) with embedding dimension 1024.

Input speech is represented as 80-dimensional log mel-filterbank coefficients computed every 10ms with a 25ms window. Global channel mean and variance normalization is applied so the trained model could be used for both streaming and offline scenarios. We set the maximum utterance duration to 75 seconds and minimum duration to 5 seconds in pre-training. During fine-tuning, a look-ahead chunk (Shi et al., 2021; Liu et al., 2021; Tang et al., 2023) is utilized and dynamic chunk training is employed by default. We alternate between offline training with infinite chunk size, and streaming training with chunk duration sampled from [160, 320, 640, 960, 1280, 1600] ms randomly within each epoch. More details about optimization, such as batch sizes and training schedulers, for different experiments are presented in Appendix B. The target labels are encoded with Sentence-Piece (Kudo & Richardson, 2018). For both speech recognition and speech translation tasks, the vocabulary is an unigram model with size 1024 and full character coverage on the corresponding training text data.

The final results are evaluated using an averaged model from checkpoints of the best 10 epochs. Speech recognition experiments are evaluated with WER while speech translation results are measure with scacre BLEU score[2]. We report both streaming and offline results. For the streaming decoding, we set the chunk size to 320ms by default if not mentioned specifically. No language model is used in all experiments.

## 4.3 FSQ MODULE

The FSQ module is built with the ResNet based encoder and decoder (Langman et al., 2024). There are 12 layers for both encoder and encoder with embedding size 512. The frontend processing is the same as the one described in §4.2. The mel-filterbank feature is with a 25ms window and 10ms shift. 4 mel-filterbank stacked features are used as input for the FSQ encoder and the reconstructed target for the FSQ decoder. The default encoder output dimension is 12 with number of levels per channel $[5, 5, 5, 5, 5, 3, 3, 3, 3, 3, 3, 3]$. It is equivalent to a codebook with vocabulary size 6,834,375.

---

[2]signature: nrefs:1|case:mixed|eff:no|tok:13a|smooth:exp|version:2.4.2

Table 1: Offline WERs for models trained on the 960 hours LibriSpeech data (WER ↓).

| Model | size(M) | dev-clean | dev-other | test-clean | test-other | ave. |
|---|---|---|---|---|---|---|
| wav2vec2 base (Baevski et al., 2020b) | 95 | 3.2 | 8.9 | 3.4 | 8.5 | 6.0 |
| wav2vec2 large (Baevski et al., 2020b) | 317 | 2.1 | 4.5 | 2.2 | 4.5 | 3.3 |
| w2v-Conformer XL (Zhang et al., 2020c) | 600 | 1.7 | 3.5 | 1.7 | 3.5 | 2.6 |
| BEST-RQ (Chiu et al., 2022) | 600 | 1.5 | 2.8 | 1.6 | 2.9 | 2.2 |
| Chunk SSL Base | 79 | 2.0 | 5.0 | 2.1 | 4.9 | 3.5 |
| Chunk SSL Large | 337 | 1.8 | 4.2 | 1.9 | 4.2 | 3.0 |

Table 2: Streaming WERs for models trained on the 960 hours LibriSpeech data (WER ↓).

| Model | size(M) | dev-clean | dev-other | test-clean | test-other | ave. |
|---|---|---|---|---|---|---|
| Conformer (Chiu et al., 2022) | 100 | 4.1 | 10.3 | 4.5 | 9.8 | 7.2 |
| Conformer (Chiu et al., 2022) | 600 | 3.9 | 9.8 | 4.4 | 9.4 | 6.9 |
| wav2vec2 (Chiu et al., 2022) | 600 | 2.7 | 8.0 | 2.9 | 7.9 | 5.4 |
| w2v-BERT (Chiu et al., 2022) | 600 | 2.7 | 8.4 | 3.0 | 8.1 | 5.6 |
| BEST-RQ (Chiu et al., 2022) | 600 | 2.5 | 6.9 | 2.8 | 6.6 | 4.7 |
| Chunk SSL Base | 79 | 2.3 | 6.6 | 2.5 | 6.4 | 4.5 |
| Chunk SSL Large | 337 | 2.1 | 5.5 | 2.3 | 5.5 | 3.9 |

## 5 EXPERIMENTAL RESULTS

### 5.1 MAIN RESULTS

The pre-trained Chunk SSL models are fine-tuned with dynamic chunk training and they could be used for both streaming and offline ASR. We list the offline and streaming recognition results from the Chunk SSL models in Table 1 and Table 2. The reported literature results are presented in the first part of the tables. It is clear that the chunk SSL pre-trained models could achieve very competitive results compared to other strong baselines. The base and large models outperform the corresponding wav2vec2 base and large models.

In the streaming evaluation, the Chunk SSL model with the base configuration excels all models reported in the literature shown in Table 2. The large configuration model pushes the WER even lower and it achieves another 0.5 average WER reduction compared to the base configure model. Comparing results in Table 1 and Table 2, the proposed method significantly reduces the performance gap between the streaming and offline applications. The WER gap between the large configure models is 0.8 in average for Chunk SSL while the average WER difference is 2.5 for BEST-RQ. Note, literature models listed in Table 1 and Table 2 are initialized with dedicated pre-trained models either offline or streaming, while a Chunk SSL model could be initialized with the same pre-trained model and operated in both modes.

In the speech translation experiments, we evaluate on both "En→Es" and "En→De" directions and the results are listed in Table 3. The baselines are initialized with a speech recognition acoustic encoder ("ASR" in the column "Init."), which is trained with the English data in the MUST-C English-Spanish direction. Both "En→Es" and "En→De" results show that the encoder initialized with Chunk SSL outperforms the one initialized with a speech recognition encoder trained with MUST-C data only. We could draw similar conclusion as the LIBRISPEECH experiment that the Chunk SSL could effectively improve speech translation quality for both streaming and offline cases.

### 5.2 IMPACT OF FSQ CODEBOOK SIZES

In order to verify our a high resolution FSQ codebook hypothesis discussed in §3, we study FSQ tokenizers with different codebook sizes. Two aspects are examined: first the agreement among FSQ tokens and phonemes, and second WERs for ASR models initialized with different FSQ tokenizers. For the agreement among FSQ tokens and phonemes, we compare the overlap of features associated with different tokens and phonemes. It can be measured by phone purity and phone-normalized mutual information (PNMI) (Hsu et al., 2021a). Phone purity denotes the conditional probability of a phoneme given a FSQ index while PNMI measures the percentage of uncertainty about a phoneme

Table 3: Speech Translation Result on the MuST-C dataset (BLEU ↑).

| Lang | Init. | Offline | | Streaming | |
|---|---|---|---|---|---|
| | | dev | tst-COMMON | dev | tst-COMMON |
| En→Es | ASR | 28.3 | 24.3 | 24.9 | 21.4 |
| | Chunk SSL | 30.3 | 26.3 | 26.8 | 23.3 |
| En→De | ASR | 21.7 | 21.8 | 18.9 | 18.5 |
| | Chunk SSL | 23.7 | 23.6 | 20.1 | 20.2 |

Table 4: Comparison of finite scalar quantization with different vocabulary sizes. "phn pur." and "PNMI" stands for phone purity and phone-normalized mutual information.

| vocab size | time(s) | levels | phn pur. | PNMI | dev-clean | dev-other |
|---|---|---|---|---|---|---|
| 1000* | 1990 | $4\,(8 \times 1 - 5 \times 3)$ | 0.19 | 0.01 | 2.1 | 5.5 |
| 1000 | 1988 | $4\,(8 \times 1 - 5 \times 3)$ | 0.19 | 0.01 | 2.2 | 5.4 |
| 50,625 | 1998 | $8\,(5 \times 4 - 3 \times 4)$ | 0.22 | 0.22 | 2.1 | 5.3 |
| 1,265,625 | 2039 | $10\,(5 \times 6 - 3 \times 4)$ | 0.67 | 0.83 | 2.0 | 5.2 |
| 6,834,375 | 2240 | $12\,(5 \times 5 - 3 \times 7)$ | 0.89 | 0.95 | 2.0 | 5.0 |
| 791,015,625 | 2250 | $14\,(5 \times 10 - 3 \times 4)$ | 0.97 | 0.99 | 2.0 | 5.4 |

label eliminated after observing a specific FSQ token. We generate phoneme based alignment with Montreal Aligner (McAuliffe et al., 2017)[3] on two LIBRISPEECH dev sets.

Table 4 lists models trained with different codebook sizes with the base configure. The first column gives FSQ vocabulary sizes, the second column provides the Chunk SSL training time (in seconds) for 1000 updates. "levels" is the FSQ levels for quantization. For example, "$4(8 \times 1 - 5 \times 3)$" in the first row means there are 4 output channels, the first channel is with level 8 and the remaining 3 channels are with level 5. They span a codebook with size 1000. "phn pur." stands for phone purity.

The first two rows listed models trained with FSQ vocabulary size 1000. "1000*" means the model is optimized with a full codebook as Eq. (4), while "1000" indicates the model is pretrained with the group masked prediction loss following Eq. (5). The results shows two models achieve comparable accuracies with similar pre-training time. As shown in Table 4, when the codebook size increases, "phn pur." and "PNMI" increase steadily. Both metrics show tokens in the codebook with a high resolution have better consistency with phonemes. This confirms our assumption that frames associated with a specific FSQ token are more likely being labelled with the same phoneme if the codebook resolution is high enough. On the other hand, we observe WER is reduced when FSQ codebook size becomes bigger. It is peaked for the codebook size around 6 millions. When the codebook size gets extremely big, i.e. 791 millions in Table 4, the "dev-other" WER increases from 5.0 to 5.4. Our hypothesis is that an extremely large codebook makes the optimization harder, and more training time might be required to get good representation. Based on above observations, we conclude that a codebook with high resolution helps Chunk SSL to transfer knowledge to the downstream task. However, an extremely large FSQ codebook might make the pre-training hard and hurt the downstream task.

The next thing we examined is computation cost. We don't observe any statistical difference for the FSQ training time when the codebook size changes, since majority of computation time is spent on data input/output, encoder and decoder Transformer/Conformer layer computation. On the other hand, a Chunk SSL training with a large codebook can increase the training time slightly that we find about 10% training time increase when the codebook size grows from 1000 to 6 millions as shown in Table 4.

## 5.3 LATENCY EVALUATION

We use the base Transducer model to analyze the impact of different chunk sizes on latency and performance. We use Length-Adaptive Average Lagging(LAAL) (Papi et al., 2022) as latency scorer,

---

[3]https://github.com/MontrealCorpusTools/mfa-models/releases/download/acoustic-english_us_arpa-v3.0.0/english_us_arpa.zip

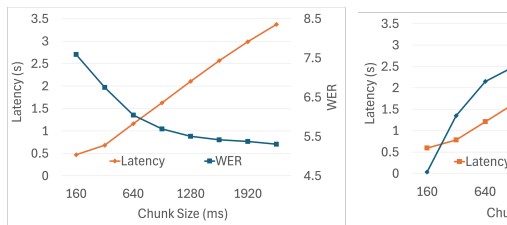 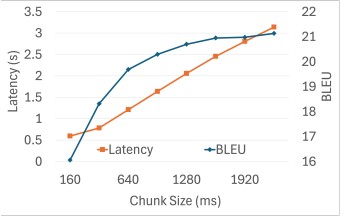 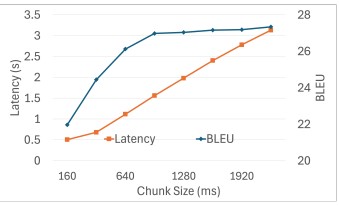

(a) LIBRISPEECH dev-other WER and Length Adaptive Average Lagging (LAAL) vs chunk size (ms)

(b) MUSTC EN-DE dev BLEU and Length Adaptive Average Lagging (LAAL) vs chunk size (ms)

(c) MUSTC EN-ES dev BLEU and Length Adaptive Average Lagging (LAAL) vs chunk size (ms)

Figure 3: Latency v.s. Performance

which is evaluated with the Simuleval toolkit (Ma et al., 2020). We employ greedy decoding instead of beam search decoding for simplicity in this study.

Figure 5.3 depicts the model performance (blue) and latency (red) under different chunk sizes for speech recognition (Figure 3a) and speech translation (Figure 3b and Figure 3c). The speech recognition is evaluated on the LIBRISPEECH dev-other set and speech translation tasks are evaluated on the corresponding MUST-C dev set. It can be seen that the model is with a small latency (LAAL 0.47 second) but high WER (7.6) when the chunk size is small (160ms). The performance improves steadily when the chunk size increases but at the cost of increasing latency. The speech translation experiments have similar observations. The results indicate the model can switch from streaming mode to offline mode smoothly as the chunk sizes change. Other latency evaluation results, such as Average Lagging (AL) (Ma et al., 2019) and Differentiable Average Lagging (DAL) (Arivazhagan et al., 2019), could be found in Appendix C

## 6 RELATED WORK

Speech self-supervised learning algorithms form contextual representations through learning to predict missing information due to time constraint or masking. van den Oord et al. (2018) propose an auto-regressive based SSL approach to predict the future samples based on the previous samples. Speech SSL methods such as wav2vec2 (Baevski et al., 2020b), HuBert (Hsu et al., 2021b) and BEST-RQ (Chiu et al., 2022) choose to randomly mask audio span and the model learns to restore those masked regions. Chunk SSL combines both approaches that it masks audio span in the extended chunk and recovers masking frame indices in an chunk based auto-regressive manner.

Vector quantization (VQ) is one of the important components in the speech SSL. Baevski et al. (2020a) leverages Gumbel-Softmax with multiple variable groups to discretize audio features. Methods such as diversity loss have to take to alleviate codeword collapse issue. Hsu et al. (2021b) resorts k-means to cluster the encoder outputs to build the codebook. In order to achieve a good representation, an iterative training is required to keep re-building the encoder with a new codebook. BEST-RQ (Chiu et al., 2022) chooses an alternative way, which initializes the codebook randomly and no further update is applied during SSL. Compared with these VQ methods aforementioned, FSQ does not suffer from codebook collapse and achieves high codebook usage (Mentzer et al., 2024).

## 7 CONCLUSIONS

In this work, a chunkwise speech self-supervised learning approach is proposed. The model is trained to learn to reconstruct the masked information in the right most chunk based on unmasked frames in the same chunk and preceding chunks. An efficient copy and append data augmentation method is proposed to parallel chunk-wise computation. FSQ is employed to generate high resolution discrete tokens and a group based decomposition method is proposed to alleviate the computation challenge. The model is trained with varied chunk durations to fit the different scenarios, hence a pre-trained model could be used for both streaming and offline applications. Our experiments show that a model initialized with Chunk SSL could achieve very competitive offline results and excellent streaming results on LIBRISPEECH and two MUST-C translation directions.

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

# A CADA PSUEDO CODE

# B OPTIMIZATION CONFIGURES

## B.1 FSQ PRE-TRAINING

The FSQ model is built with learning rate is 2.0e-4 and ExponentialLR scheduler from PyTorch. The effective batch size is 3200 seconds with up to 180,000 updates. The FSQ module is optimized with MSE loss to restore mel-filterbank features from their quantized audio representations. It takes 4 A10 GPUs for 42 hours to build a FSQ model.

## B.2 CHUNK SSL PRE-TRAINING SETTING

The pre-training is scheduled as Transformer (Vaswani et al., 2017) with warmup step 25,000 and maximum learning rate 3e-4. The batch size is 16,000 seconds for the base configure model. We choose a smaller effective batch size for the large configure model with 6,000 seconds due to computation resource limitation. The model is trained up to 400,000 updates. It takes about 10 days for 8 A100 GPUs to build a Chunk SSL encoder with the base configure.

## B.3 LIBRISPEECH FINE-TUNING SETTING

**960 hours fine-tuning** For the base configure model, we choose the three stages scheduler scheme as (Baevski et al., 2020b), i.e., warmup, hold and annealing. The warmup stage takes about 1000 updates with encoder parameters frozen, the hold stage takes about 40,000 updates and the remaining 60,000 updates are in an annealing stage. The peak learning rate in fine-tuning is 2.0e-3. The SpecAugment (Park et al., 2019) data augmentation is applied with 2 frequency maskings of 27, and 10 time maskings at a maximum ratio of 0.05 in fine-tuning experiments. The effective batch size for fine-tuning is 10,000 seconds. For the large configure model, we choose Transformer scheduler (Vaswani et al., 2017) since it has less hype-parameters to explore. The warmup step is 5000 with encoder parameters frozen. Similar as the base configure model, the total update step is up to 100,000. The peak learning rate is 5e-4. The effective batch size for the large model is 7200 seconds.

---

**Algorithm 1** CADA Computation With Attention and Convolution Modules.

---

1: **procedure** CADA_AUGMENT($X, C, \text{mask\_rate}$)
2: $\quad M \leftarrow \lfloor N/C \rfloor, N_{\text{copy}} \leftarrow N - C, N_{\text{aug}} \leftarrow N + N_{\text{copy}}$
3: $\quad \text{mask} \leftarrow \mathbf{0}^{N_{\text{copy}}}$
4: $\quad$**for** copied_idx $= 1$ to $M - 1$ **do**
5: $\quad\quad \text{local\_start} \leftarrow (\text{copied\_idx} - 1)C, \text{span\_len} \leftarrow \lfloor C \cdot \text{mask\_rate} \rfloor$
6: $\quad\quad \text{span\_start} \leftarrow \text{random\_integer}(\text{local\_start}, \text{local\_start} + C - \text{span\_len})$
7: $\quad\quad$**for** $t = \text{span\_start}$ to $\text{span\_start} + \text{span\_len} - 1$ **do**
8: $\quad\quad\quad \text{mask}[t] \leftarrow 1$
9: $\quad\quad$**end for**
10: $\quad$**end for**
11: $\quad X_{\text{aug}} \leftarrow \text{concatenate}(X[0 : C * M], X[C : (M - 1)C]), X \leftarrow X_{\text{aug}}$
12: $\quad$**for** $t = 0$ to $N_{\text{copy}} - 1$ **do**
13: $\quad\quad$**if** $\text{mask}[t] = 1$ **then**
14: $\quad\quad\quad X[N + t] \leftarrow 0$
15: $\quad\quad$**end if**
16: $\quad$**end for**
17: $\quad \alpha \leftarrow \text{init\_attn\_mask}(N_{\text{aug}}, N_{\text{aug}}, C)$
18: $\quad$**for** $\ell = 1$ to $L$ **do**
19: $\quad\quad Q, K, V \leftarrow W_q^\ell X, W_k^\ell X, W_v^\ell X$
20: $\quad\quad$**for** $i = 0$ to $N_{\text{aug}} - 1$ **do**
21: $\quad\quad\quad \text{max\_score} \leftarrow \max_j(\beta Q[i] \cdot K[j] - \zeta(1 - \alpha[i, j]))$
22: $\quad\quad\quad s[j] \leftarrow \exp(\beta Q[i] \cdot K[j] - \zeta(1 - \alpha[i, j]) - \text{max\_score}), \text{denom} \leftarrow \sum_j s[j]$
23: $\quad\quad\quad A[i, j] \leftarrow s[j]/\text{denom}, X_{\text{att}}[i] \leftarrow \sum_j A[i, j]V[j]$
24: $\quad\quad$**end for**
25: $\quad\quad X \leftarrow X_{\text{att}}$
26: $\quad\quad X_{\text{new}} \leftarrow \text{zeros\_like}(X)$
27: $\quad\quad$**for** $ci = 0$ to $M - 1$ **do**
28: $\quad\quad\quad s, e \leftarrow ciC, ciC + C, X_{\text{seg}} \leftarrow \text{concatenate}(X[\max(s - L_{\text{lc}}, 0) : s], X[s : e], X[N + s : N + e] + PAD)$
29: $\quad\quad\quad X_{\text{conv}} \leftarrow \text{Convolution}_\ell(X_{\text{seg}})$
30: $\quad\quad\quad X_{\text{new}}[s : e] \leftarrow X_{\text{conv}}[0 : C], X_{\text{new}}[N + s : N + e] \leftarrow X_{\text{conv}}[C : 2C]$
31: $\quad\quad$**end for**
32: $\quad\quad X \leftarrow X_{\text{new}}$
33: $\quad$**end for**
34: $\quad \text{loss,count} \leftarrow 0, 0, O \leftarrow X[N :]$
35: $\quad$**for** $t = 0$ to $N_{\text{copy}} - 1$ **do**
36: $\quad\quad$**if** $\text{mask}[t] = 1$ **then**
37: $\quad\quad\quad o_i, \mu_i \leftarrow O[t], \text{group\_FSQ\_labels}[t]$
38: $\quad\quad\quad$**for** $r = 1$ to $R$ **do**
39: $\quad\quad\quad\quad e_r \leftarrow \text{embedding}_r[\mu_i[r]], \text{loss} \leftarrow \text{loss} - \log \frac{\exp(o_i \cdot e_r)}{\sum_j \exp(o_i \cdot \text{embedding}_r[j])}$
40: $\quad\quad\quad$**end for**
41: $\quad\quad\quad \text{count} \leftarrow \text{count} + 1$
42: $\quad\quad$**end if**
43: $\quad$**end for**
44: $\quad \text{loss} \leftarrow \text{loss}/\text{count}$
45: $\quad$**return** $X, \text{loss}$
46: **end procedure**

---

**10 hours fine-tuning** We present the low resource results with 10 hours fine-tuning data at Appendix D. The model is optimized with a two-stage scheduler with warmup and hold stages. The warmup and hold steps are 2000 and 8000 for both base and large configures. The encoder is frozen during the warmup stage. Small peak learning rate 8e-05 and 3e-5 are used for the large and base models respectively. We use different SpecAugment parameters for the 10 hours fine-tuning with 1 frequency maskings of 34, and 10 time maskings at a maximum ratio of 0.02.

## B.4 MUST-C FINE-TUNING SETTINGS

The translation Transducer model is optimized with Transformer (Vaswani et al., 2017) scheduler with warmup step 10,000 and total update step up to 100,000. The peak learning rate is 5e-4. The effective batch size is 6400 seconds.

## C LATENCY RESULTS

Table 5: Latency and WER vs chunk size on base model trained with LIBRISPEECH evaluated on `dev-other`.

| Chunk Size (ms) | WER | LAAL | AL | AP | DAL | ATD |
|---|---|---|---|---|---|---|
| 160 | 7.6 | 469.1 | -1753.6 | 1.0 | 900.1 | 760.6 |
| 320 | 6.8 | 682.6 | -1492.2 | 1.1 | 1133.9 | 908.5 |
| 640 | 6.0 | 1161.3 | -833.6 | 1.2 | 1686.8 | 724.9 |
| 960 | 5.7 | 1627.8 | -177.9 | 1.3 | 2223.4 | 785.5 |
| 1280 | 5.5 | 2106.2 | 492.3 | 1.4 | 2754.8 | 928.1 |
| 1600 | 5.4 | 2566.1 | 1116.8 | 1.5 | 3246.2 | 1083.3 |
| 1920 | 5.4 | 2988.4 | 1686.7 | 1.5 | 3678.4 | 1230.6 |
| 2240 | 5.3 | 3373.7 | 2215.2 | 1.6 | 4058.9 | 1365.1 |

Table 6: Latency and BLEU vs chunk size on base model trained with MUSTC EN-DE `dev` dataset.

| Chunk Size (ms) | BLEU | LAAL | AL | AP | DAL | ATD |
|---|---|---|---|---|---|---|
| 320 | 18.3 | 782.0 | -2475.3 | 1.4 | 1329.9 | 447.8 |
| 640 | 19.7 | 1208.5 | -1841.5 | 1.6 | 1826.4 | 440.0 |
| 960 | 20.3 | 1635.9 | -1188.3 | 1.7 | 2308.0 | 538.9 |
| 1280 | 20.7 | 2056.2 | -534.5 | 1.8 | 2779.6 | 665.1 |
| 1600 | 20.9 | 2455.6 | 91.6 | 1.8 | 3190.3 | 781.8 |
| 1920 | 21.0 | 2803.6 | 625.1 | 1.9 | 3558.3 | 904.7 |
| 2240 | 21.1 | 3137.0 | 1141.6 | 1.9 | 3882.8 | 1010.6 |

Table 7: Latency and BLEU vs chunk size on base model trained with MUSTC EN-ES `dev` dataset.

| Chunk Size (ms) | BLEU | LAAL | AL | AP | DAL | ATD |
|---|---|---|---|---|---|---|
| 320 | 24.4 | 680.4 | -2601.2 | 1.3 | 1277.3 | 495.7 |
| 640 | 26.1 | 1115.2 | -1905.6 | 1.5 | 1791.4 | 461.7 |
| 960 | 27.0 | 1561.8 | -1248.4 | 1.6 | 2297.2 | 559.9 |
| 1280 | 27.0 | 1979.6 | -607.8 | 1.7 | 2782.1 | 680.3 |
| 1600 | 27.2 | 2402.3 | 33.2 | 1.8 | 3228.4 | 819.5 |
| 1920 | 27.2 | 2781.1 | 584.0 | 1.8 | 3630.6 | 940.3 |
| 2240 | 27.3 | 3131.4 | 1106.7 | 1.9 | 3990.4 | 1047.7 |

The full latency evaluation results including Average Lagging (AL) (Ma et al., 2019), Length Adaptive Average Lagging (LAAL) (Papi et al., 2022), Average Proportion (AP) (Cho & Esipova, 2016), Differentiable Average Lagging (DAL) (Arivazhagan et al., 2019), and Average Token Delay (ATD) (Kano et al., 2023) for the LIBRISPEECH and MUST-C are presented in Table 5, Table 6 and Table 7.

Table 8: WERs for model trained with 10 hours LIBRISPEECH data.

| Model | dev-clean | dev-other | test-clean | test-other | ave. |
|---|---|---|---|---|---|
| wav2vec2 Base (Baevski et al., 2020b) | 10.9 | 17.4 | 11.1 | 17.6 | 14.3 |
| wav2vec2 Large (Baevski et al., 2020b) | 6.3 | 9.8 | 6.3 | 10.0 | 8.1 |
| Chunk SSL Base (O) | 10.9 | 18.6 | 10.9 | 19.2 | 14.9 |
| Chunk SSL Base (S) | 12.0 | 21.8 | 12.0 | 22.3 | 17.0 |
| Chunk SSL Large (O) | 9.6 | 15.3 | 10.0 | 15.9 | 12.7 |
| Chunk SSL Large (S) | 10.5 | 18.5 | 10.9 | 18.9 | 14.7 |

Table 9: WERs for models trained on the 960 hours LibriSpeech data with different initialization.

| Model | Mode | dev-clean | dev-other | test-clean | test-other | ave. |
|---|---|---|---|---|---|---|
| Random | Offline | 2.3 | 5.8 | 2.4 | 5.8 | 4.1 |
| Chunk SSL + BEST-RQ | Offline | 2.1 | 5.3 | 2.2 | 5.3 | 3.7 |
| Chunk SSL + FSQ (50k) | Offline | 2.1 | 5.3 | 2.2 | 5.1 | 3.7 |
| Chunk SSL + FSQ (6m) | Offline | 2.0 | 5.0 | 2.1 | 4.9 | 3.5 |
| Random | Streaming | 2.6 | 7.4 | 2.8 | 7.2 | 5.0 |
| Chunk SSL + BEST-RQ | Streaming | 2.4 | 6.8 | 2.7 | 6.8 | 4.6 |
| Chunk SSL + FSQ (50k) | Streaming | 2.4 | 6.9 | 2.5 | 6.7 | 4.6 |
| Chunk SSL + FSQ (6m) | Streaming | 2.3 | 6.6 | 2.5 | 6.4 | 4.5 |

## D  LOW RESOURCE

The self-supervised trained model can leverage large amounts of unlabeled data to build an universal representation, which makes it possible to build a speech recognition model with relatively small amounts of labeled data. We examine Chunk SSL pre-trained models with 10 hours LIBRISPEECH data (Kahn et al., 2019). We chose a small predictor, which is with input embedding size 128 and forward layer dimension 512, due to limited amount of data. The results are demonstrated in Table 8, where "O" and "S" stand for offline and streaming recognition respectively. According to Table 8, Chunk SSL models achieve compelling results even with just 10 hours of data. The base configure model obtains similar or slightly better offline results compared with the wav2vec2 base model in the two clean datasets, and about 1 WER worse for the two other datasets. The Chunk SSL large model reduces the WER further compared with the base configure model, though they are still significant behind the corresponding wav2vec2 large model. We believe it is due to limited computation resource for the large model pre-training. We have to choose a smaller batch size for the large model and re-use the same configure from the base model pre-training. On the other hand, the streaming results are promising that the streaming model only increase the WER by less than 20% compared with the corresponding offline results. It shows the effectiveness of the proposed Chunk SSL pre-training.

## E  COMPARISON FSQ V.S. BEST-RQ

In this study, we compare the speech recognition performance with different initialization approaches: 1) random initialization ("Random"), 2) Chunk SSL model with discretized token created by BEST-RQ (Chiu et al., 2022) ("Chunk SSL + BEST-RQ"), 3) Chunk SSL model with FSQ codebook size 50,625 ("Chunk SSL + FSQ (50k)") and 4) Chunk SSL model with FSQ codebook size 6,834,375 ("Chunk SSL + FSQ (6m)"). The results are listed in Table 9.

