# OpenReview forum: "Chunk Based Speech Pre-training with High Resolution Finite Scalar Quantization"
_ICLR.cc/2026/Conference — Submitted to ICLR 2026_

### Official Review · Reviewer_scQb · 2025-11-02

**Soundness:** 3
**Presentation:** 3
**Contribution:** 2
**Rating:** 2
**Confidence:** 4

**Summary:**

This paper proposes a chunk-based self-supervised learning (SSL) framework that masks both future chunk information and sampled frames within each chunk. The method is efficiently implemented through copy-and-append data augmentation techniques, enabling scalable training. In addition, the paper introduces a multi-label prediction scheme, inspired by audio codecs, to efficiently handle large vocabulary sizes. Training is conducted using dynamic chunk sizes, allowing the model to learn representations across varying temporal contexts, while inference leverages the chunk structure to balance latency and performance. Experimental results on LibriSpeech ASR and MuST-C speech translation tasks demonstrate the effectiveness of the proposed approach.

**Strengths:**

- The paper presents an efficient implementation of chunk-based SSL, which is particularly valuable for streaming ASR and ST applications where low-latency processing is critical.
- The paper also provides an efficient discretization by using multi-label classification.
- The proposed approach demonstrates strong experimental performance, outperforming existing SSL-based methods on both ASR and ST benchmarks.

**Weaknesses:**

* **Unclear masking implementation (potentially critical):**
  The manuscript’s description of the masking operation is ambiguous. The text appears to apply masks by multiplying masked positions by **0** rather than using large negative values before the softmax (or otherwise excluding them from the attention denominator). If true, this would leave nonzero attention weights on masked/future positions (the softmax denominator still includes those entries), allowing the model to attend to information that should be hidden. This is a fundamental concern that could invalidate the reported results. Please clarify precisely how masking is implemented (include equations), and provide an ablation that compares the current implementation with a standard masked-attention formulation (e.g., add −∞ or a large negative constant before softmax) to show the effect on performance and verify correctness.

* **Limited novelty and missing literature context for multi-label representations:**
  The multi-label / codec-like representation component echoes several recent works on discrete and multi-label audio representations (for example, MMM and other codec-inspired SSL approaches). The paper currently does not situate its contribution clearly within this literature nor convincingly establish novelty. Please expand the related-work discussion to cover recent multi-label and codec-based representation studies, and clarify what is new here (algorithmic change, efficiency, empirical regime, etc.). If novelty is incremental, the manuscript should present stronger empirical or analytical evidence that the proposed variant offers a meaningful advantage over prior methods.
  * Shi, Jiatong, et al. "MMM: Multi-Layer Multi-Residual Multi-Stream Discrete Speech Representation from Self-supervised Learning Model." Proc. Interspeech 2024. 2024.

* **Insufficient analysis of representation trade-offs (phoneme purity alone is inadequate):**
  The analysis relies heavily on phoneme-label purity as a measure of representational quality. However, phoneme purity alone favors larger codebook / vocabulary sizes and does not capture clustering compactness or redundancy trade-offs. The paper should include additional probing analyses based on **cluster purity**, and report trade-offs between vocabulary size, cluster purity, and downstream performance. Presenting such metrics (and plotting cluster/phoneme purity for each vocabulary size) would provide a more trustworthy and nuanced interpretation of why larger vocabularies help or hurt.

**Questions:**

My primary questions are already listed in the Weaknesses field. If these questions are addressed satisfactorily, I am open to revising my scores.

I also have a few minor questions and suggestions that could improve the clarity and comparability of the paper:

- Why was Conformer chosen as the backbone? Many SSL models (e.g., Wav2Vec 2.0, HuBERT, WavLM) are based on standard Transformers, which would allow for a clearer comparison.
- Similarly, regarding model size, it would improve comparability if the proposed method used a similar number of parameters as the baseline models.

**Details Of Ethics Concerns:**

This is not the authors’ fault, but I would like to note an important ethical and legal concern. The use of the TED corpus for AI development is explicitly prohibited under TED’s Terms of Use (see Section 6, License to TED Content: https://www.ted.com/about/our-organization/our-policies-terms/ted-com-terms-of-use). Consequently, the paper should avoid using the MuST-C dataset, which is derived from TED talks. This issue is widely recognized in the speech research community, and for the same reason, the IWSLT challenges have discontinued the use of the TED corpus. It would be advisable for the authors to clarify their data usage and, if possible, replace or supplement MuST-C with legally unambiguous datasets.

It is worth noting that some papers still report results using TED-based datasets (e.g., HF Open ASR leaderboard, https://huggingface.co/spaces/hf-audio/open_asr_leaderboard). I personally do not believe this should penalize the paper, given current community conventions. This comment is intended as a caution for potential future risks and a suggestion that the authors consider replacing TED-based data with alternative, legally unambiguous datasets where possible.

---

> ### Author Response · Authors · 2025-11-20
> **Rebuttal By Authors**
>
> We thanks reviewer scQb for pointing out issues in the manuscripts and we will fix the typo issues in the revised version.
>
> `masks by multiplying masked positions by 0 rather than using large negative values`
>
> Thanks point out this. There is a typo in Equ 1.  The masked attention matrix is used to select component to fill with a large negative number. The updated Equ. 5 is
>
> $A_{i,j} = \frac{\exp\big(\beta (W_q x_i)^T (W_k x_j) - \zeta (1 - \alpha_{i,j})\big)}{\sum_{\tau} \exp\big(\beta (W_q x_i)^T (W_k x_\tau) - \zeta (1 - \alpha_{i,\tau})\big)}$
>
> where $\zeta$ is a big number, such as 1e6.
>
>
>
> `Limited novelty and missing literature context for multi-label representations`
> The main contributions of this work have two-fold. First, a chunk based SSL training is proposed to unified streaming and offline pre-training. As far as we know, it is the first work to unify streaming and offline speech pre-training.
> The results show good improvement achieved for the streaming case while strong offline performance is preserved. Second, a high resolution FSQ
> based quantization approach is propsed to further improve the performance. Our experiments show that FSQ is no better than other quantization approaches such as BEST-RQ. However, when we increase the codebook size to a few millions, the proposed quantization method would help to boost the downstream task further.  It emphasizes the importance of the high resolution FSQ codebook and our group based FSQ loss computation.
>
> In MMM[1], RVQ based codecs and codecs from different layers have been studied. However, it is challenging to extend them for masking based pre-training. RVQ or multilayer based codecs has strong dependency among groups, we can't decompose them as we did to FSQ token. Or in other words, we can't use a high resolution codebook up a few millions for the pre-training, which is critical to achieve good performance on the downstream tasks.
> We will add this discussion in our revision.
>
> `Insufficient analysis of representation trade-offs for cluster purity`
> Our hypothesis is that if we could conduct a pre-training task with fine-grained target labels compared with the downstream task, it will boost the downstream task performance. For example, the pre-training tasks has 6 labels, p1, p2, p3, ..., and p6. The downstream task has 2 labels d1 and d2. If most samples to p1, p2, p3, p4 are associated with d1, while samples for p5 and p6 are with d2. The downstream task can leverage
> pre-training model efficiently since it just need to learn to map those frames p1 to p4 to the new label d1 and others to d2. The hypothesis is
> proved in Table 4. It clearly shows the large codebook size until a few millions could boost the downstream ASR performance. We don't list cluster purity number since it always penalize the codecs with a large codebook size and not relevant to our hypothesis.
>
>
> `Why was Conformer chosen as the backbone? Many SSL models (e.g., Wav2Vec 2.0, HuBERT, WavLM) are based on standard Transformers, which would allow for a clearer comparison. Similarly, regarding model size, it would improve comparability if the proposed method used a similar number of parameters as the baseline models.`
> Conformer gains more popularity against the transformer framework and it is the main reason we choose Conformer as backbone. Another reason is it
> is very expensive to conduct pre-training experiments and we have limited ourself to Conformer only due to resource constraints. I understand reviewer's concerns about fair comparison. We tried our best to reproduce a BEST-RQ based system for fair comparison as below.
> In the table, we list base configuration systems with different initialization. The *Random'* means no pre-training is applied. The second and third rows are models initialized with pre-trained models with BEST-RQ and FSQ with moderate codebook sizes, i.e. 8k and 50k respectively. The results show there is no big difference for BEST-RQ and FSQ if the codebook size is moderate and FSQ could outperform all initializations if the codebook size increases to a few millions (6m).  We will include the new results in our revision.
>
> | Model                     | dev-clean | dev-other | test-clean | test-other | ave. |
> |---------------------------|-----------|-----------|------------|------------|------|
> | Random                    | 2.3       | 5.8       | 2.4        | 5.8        | 4.1  |
> | Chunk SSL + BEST-RQ (8k)  | 2.1       | 5.3       | 2.2        | 5.3        | 3.7  |
> | Chunk SSL + FSQ (50k)     | 2.1       | 5.3       | 2.2        | 5.1        | 3.7  |
> | Chunk SSL + FSQ (6m)      | 2.0       | 5.0       | 2.1        | 4.9        |  3.5 |
>
>
>
> [1] Shi, Jiatong, et al. "MMM: Multi-Layer Multi-Residual Multi-Stream Discrete Speech Representation from Self-supervised Learning Model." Proc. Interspeech 2024.

---

### Official Review · Reviewer_kQdN · 2025-11-04

**Soundness:** 3
**Presentation:** 2
**Contribution:** 3
**Rating:** 6
**Confidence:** 4

**Summary:**

The paper proposes Chunk SSL, a chunk-wise self-supervised pretraining framework that aims to unify streaming and offline speech encoders. Core pieces: (1) Copy-and-Append Data Augmentation (CADA) to parallelize chunked masked-prediction pretraining; (2) Finite Scalar Quantization (FSQ) to discretize frames, with very large codebooks; and (3) a group masked-prediction loss to keep compute/memory manageable with high-resolution codebooks.

**Strengths:**

- The paper proposes an alternative streaming SSL recipe that aims to unify pretraining for both streaming and offline speech encoders within a single framework

**Weaknesses:**

- The method largely recombines familiar pieces—chunked SSL, discretization, and masking—with the main novelty in CADA and the FSQ + group-loss design.
- Given how widely SSL embeddings are used, the paper’s downstream evaluation feels narrow; broader tasks (e.g. speaker verification/diarization) would better demonstrate generality.
- Although effective, the approach depends on millions-level FSQ codebooks; as the vocabulary grows, optimization becomes harder and WER degrades—raising concerns about training stability and compute cost.
- Some implementation details for CADA are hard to follow; it would help to include a concise algorithm box/pseudocode.

**Questions:**

- I see clear gains for streaming ASR, but the offline WERs trail the comparative approach. Could you explain why? Intuitively, when the streaming constraints are relaxed, an effective method should at least match—ideally surpass—offline baselines.
- As a follow-up: if CADA is strictly equivalent to naïve chunk SSL (only enabling parallelization), does that imply the conventional naïve approach could also support streaming—and, if trained identically, would its offline performance remain competitive?
- Please also address the weaknesses noted above. I’m happy to raise my score if these concerns are satisfactorily resolved.

---

> ### Author Response · Authors · 2025-11-20
> **Rebuttal By Authors**
>
> Thank you reviewer kQdN for your positive feedback and valuable suggestions. We include the CADA pseudo code in [post](https://openreview.net/forum?id=ISSxXXiu3w&noteId=gIe41MgToK) as you suggested and address your questions and concerns as below:
>
> Q1`I see clear gains for streaming ASR, but the offline WERs trail the comparative approach. Could you explain why? `
>
> As show in table 1, wav2vec2's base model achieved an average score of 6.0 on librispeech, whereas our base model achieved an average score of 3.5 despite having fewer parameters (wav2vec2 base 95 MB and chunk SLL has 79 MB). The wav2vec2 large model achieved an average score of 3.3 and our large model achieved an aveage score of 3.0. In both offline cases where the model size is comparable, our offline number actually achieved a better result than other SSL methods.
> Compared with BEST-RQ large model, the large Chunk SSL model is slightly worse. There are two main reasons the Chunk SSL model is behind.
> First the Chunk SSL large model is only half the size of the corresponding BEST-RQ large model. Second, BEST-RQ model is optimized with the utterance based channel mean normalization since it is an offline model while the Chunk SSL model doesn't conduct this normalization due to streaming constraints. In our study, we found there is about 0.1 to 0.2 WER degradation in the Librispeech dataset due to the channel mean normalization difference.
>
> Q2 `does that imply the conventional naïve approach could also support streaming—and, if trained identically, would its offline performance remain competitive?`
>
> Please let us try to clarify the method of naive chunk SSL. What we mean is that if we were to train the model to predict future frames, a naive way would be to mask all future frames and only present an utterance of chunk 1 to the model. The model will be expected to predict future chunk 2. In the next iteration we present data in chunks 1 and 2, and the model will be expected to predict future chunk 3. So we present chunks 1 to N-1 and the model expected to predict chunk N. So for each utterance we will need to repeat the utterance with N-1 chunks in order to train the model to predict the future chunk
>
> CADA not only enables parallelism during training, but it also enables us to use lookahead chunk during inference time. Prior work includes EmFormer[1]. The model supports both streaming and offline modes in one model. However, the work hasn't ever been extended to self supervised learning.
>
>
> `as the vocabulary grows, optimization becomes harder and WER degrades—raising concerns about training stability and compute cost.`
>
> The computational issue associated with high FSQ codebook size can be alleviated by using group masked prediction loss. By decomposing the codebook into groups of subcodebooks, we can drastically decrease the storage requirement. As demonstrated in our paper section 3.2, with a vocabulary size 1,2,65,625 and embedding size 512, it would take about 2.4 GB whereas using group prediction loss this number can be lowered to 84KB. Table 4 also shows that large codebooks can achieve better results with small increases in training time. In our experiments, the training FSQ model or the pre-training based the FSQ model is very stable. Though, we do observe a FSQ model with extremely large codebook might make the pre-training optimization hard. The conclusion is that we don't need to worry about computation cost or training stability for the FSQ model with large codebook.
>
> ` the paper’s downstream evaluation feels narrow; broader tasks (e.g. speaker verification/diarization) would better demonstrate generality.`
>
> Thanks for the suggestions. The main focus of our work is to unify the pre-training for both streaming and offline applications. So we just need to conduct pre-training once and build one model for both scenarios instead of two pre-trainings for two pre-trained models. It would be interesting to evaluate the model on other downstream tasks such as speaker verification/diarization, it will be our future work.
>
> [1] Shi, Yangyang et el. "Emformer: Efficient memory transformer based acoustic model for low latency streaming speech recognition." ICASSP (2021)

---

### Official Review · Reviewer_Ve4f · 2025-11-05

**Soundness:** 3
**Presentation:** 2
**Contribution:** 2
**Rating:** 4
**Confidence:** 5

**Summary:**

The paper proposes an integrated framework for both non-streaming and streaming self-supervised speech training. By introducing a new CADA masking strategy and FSQ-based target prediction, the unified model achieves performance comparable to prior specialized models for either setting. However, additional ablation studies are required to justify the benefit and usage of CADA and FSQ. In particular, CADA might be replicable by simply applying both non-causal and block-wise causal attention masks during self-supervised training, without the need for copy and append. Likewise, a comparison for adopting FSQ over continuous targets would greatly justify its use.

**Strengths:**

1. The paper introduces a model capable of dynamically adjusting the chunk size during both training and inference, enabling users to flexibly control latency and reducing the mismatch between training and inference.

2. The proposed CADA approach might be a potential alternative to previous masking strategies for self-supervised prediction, e.g. Best-RQ and Wav2vec 2.0.

**Weaknesses:**

1. If the goal is to obtain fine-grained prediction targets, why not consider continuous representation could serve as a more efficient alternative, e.g. mel-filterbank features. The benefit of quantizing mel-filterbank features using FSQ for mask prediction is not clearly established. Since FSQ quantizes each channel independently, it is reminiscent of the approach in dMel: Speech Tokenization Made Simple. However, it seems plausible that continuous mel-filterbank features could suffice for training. A direct comparison between models predicting continuous mel-filterbank features and those using FSQ targets would be necessary to justify the added complexity of introducing an FSQ encoder.

2. As shown in Figure 2(b), it is unclear why the “1st chunk” is not appended as well. Assigning zero attention scores to the 'full context 1st chunk' to the 'appended 1st chunk' should work.

3. Following above, when using “offline training with infinite chunk size", Figure 2 shows no indication of the masked chunk being copied and appended for prediction, which does not seem right.

4. A more thorough comparison with simple joint causal and non-causal masking during training would strengthen the justification for using CADA. For example, training a Best-RQ-style alternative that includes both chunk-wise causal attention mask and non-causal attention masks within each batch would be a fair baseline. This could also justify CADA using twice the computation during training.

**Questions:**

1. What is the motivation for introducing an additional FSQ module for target prediction instead of directly using mel-filterbank features?

2. In the non-streaming training setup, how is masked prediction performed if the first chunk is not appended?

3. Have you explored training a unified model for both streaming and non-streaming scenarios without the copy and append operation?

---

> ### Author Response · Authors · 2025-11-20
> **Rebuttal By Authors**
>
> Thank you reviewer Ve4f for your spending time to give us valuable feedback. Below are our replies for your comments and questions. We also include the CADA pseudo code in the previous [post](https://openreview.net/forum?id=ISSxXXiu3w&noteId=gIe41MgToK). Please let us know if you have any further questions.
>
> Q[1] `The motivation for introducing an additional FSQ module for target prediction instead of directly using mel-filterbank features`
>
> That is a great question. Yes, using mel-filterbank features are targets has been explored by previous speech SSL algorithms[1]. The main purpose of the pre-training is to initialize the model with large amount of unlabeled data to generate contextual representations. In the masking based pre-training scheme, the model learns to construct the information by predicting the labels with help of contextual information from unmasked frames. If we use continuous features directly as target, the encoder output is mel-filterbank, which only contains short span information, instead of contextual representation beneficial for the downstream tasks. That is main reason we choose discretized unit instead of continuous feature representation as our target.
>
> Q[2,3] `How is masked prediction performed if the first chunk is not appended? Have you explored training a unified model for both streaming and non-streaming scenarios without the copy and append operation?`
>
> Our training setup is used to pre-train model for both streaming and non-streaming. We do not build model for an offline only case. From both literature and our study has shown that dynamic chunk training works well for both streaming and offline scenarios. There are other models that try to unified streaming and offline mode. But none have been done in self-supervised learning. Yu et al [2] propose to use different attention masks, convolution masks and pooling for streaming and offline self-supervised learning. Due to this discrepancy, its conformer result the librispeech dev other degraded from 5.0 to 5.9 for a dual mode model compared to a standalone one trained just for offline mode.
>
> [1] Kawakami, Kazuya et al. “Learning Robust and Multilingual Speech Representations.” Findings of EMNLP 2020.
> [2] Yu, Jiahui et al. “Dual-mode ASR: Unify and Improve Streaming ASR with Full-context Modeling.” ICLR (2020).

---

### Official Review · Reviewer_TKjS · 2025-11-07

**Soundness:** 2
**Presentation:** 1
**Contribution:** 2
**Rating:** 2
**Confidence:** 4

**Summary:**

This work proposes a chunkwise pretraining scheme for speech self-supervised learning (SSL), aiming to enhance streaming capability for downstream tasks, including automatic speech recognition (ASR) and speech-to-text translation (ST). The authors develop a training scheme that allows efficient training of the chunkwise speech SSL by using the specialized "copy and append data augmentation (CADA)" technique. In addition, they employ a masked-prediction objective using finite scalar quantization (FSQ) codes as targets. They evaluate their Chunk SSL speech models on ASR and ST, reporting both offline (non-streaming) and online (streaming) performance.

**Strengths:**

[S1]: The motivation is clear. The chunkwise SSL could intuitively help on the streaming performance of the specified downstream tasks.

[S2]: The method appears to be training-efficient.

**Weaknesses:**

[W1]: The presentation requires further refinement. I cannot fully grasp the precise meaning of some sentences. Even the context of the core design—CADA—is somewhat confusing. See [Q1].

[W2]: The completeness and soundness of the experiments are questionable. The authors propose CADA and FSQ on speech SSL, but I cannot infer how effective CADA or FSQ are individually from the provided results. See [Q2].

[W3]: The novelty is limited. The novelty mainly lies in CADA. However, I consider CADA as a technique for efficiently train the chunkwise SSL model, at the cost of increased memory usage due to the "copy-and-append" behavior. The idea of trading memory for time is not particularly novel or impressive. I also have a few questions regarding the core design choices; please see [Q3].

**Questions:**

**[Q1] Questions regarding the presentation:**

[Q1-1] In L74-76, *"We hypothesize that speech frames associated with a high resolution FSQ token are mainly mapped to an unique modeling unit in the downstream task and it makes the knowledge transfer easier from the pre-training stage to the fine-tuning stage."*
What does this sentence mean? What is *"an unique modeling unit in the downstream task"*? Do you have any support on this?

[Q1-2] In L106-109, *"Inspired by the implementation of the chunk encoder with a look-ahead chunk for the streaming speech translation (Liu et al., 2021), we propose a copy and append data augmentation for the Chunk SSL, which reorganizes input data and changes the augmented data computation, but it is still strictly equivalent to the computation in the naive Chunk SSL algorithm."*
I am unsure which part of Liu et al.’s work inspired you, and what exactly the ‘naive chunk SSL algorithm’ refers to (seems to be a critical baseline).

[Q1-3] In L115-118, *"Assuming the chunk size is 4 frames, the speech features are segmented into three base chunks (in cyan): “B1”,
“B2” and “B3”. Chunk “B2” and “B3” are copied and appended to the end of the utterance as “E1” and “E2” (in lime). They correspond to the extended chunks of “B1” and “B2” respectively. There is no extended chunk for the last chunk (“B3”)."*
The statement that chunks “B2” and “B3” are copied and appended (extended) contradicts the claim that there is no extended chunk for the last chunk “B3.”

[Q1-4] Overall, I cannot quite get why we need to ‘copy and append,’ and how intuitively effective this approach can be, given that the *naive chunk SSL algorithm* is not well introduced or clearly defined as a baseline.

[Q1-5] In L135, what is X? You should define the transformation from X to X′ (the CADA-augmented utterance) more precisely and carefully.

[Q1-6] Figure 2-(b) is confusing. In the main text that discuss it, why does "B1" pair with "E1" (L203)? Shouldn’t "B2" pair with "E1," according to Figure 1?

---

**[Q2] Questions about the soundness of experiments**

[Q2-1] In Table 1 and Table 2, only the Chunk SSL scores are reported. However, to demonstrate the effectiveness of the proposed method (CADA), the so-called *"naive Chunk SSL"* should be included as a baseline.

[Q2-2] Following the previous question, how much benefit is gained from using FSQ?

[Q2-3] Why does Table 3 not include other SSL methods?

[Q2-4] Are all baselines in Tables 1 and 2 taken from the literature? If so, how do you ensure the consistency of experiments between the previous work and your work? Has BEST-RQ released its codebase?

---

**[Q3] Questions regarding the core design of CADA**

[Q3-1] Can you elaborate on the novelty of CADA? What do you think is beyond "technical training efficiency improvements"?

[Q3-2] What is "copy and append" mainly about? Is it about 1. enabling look ahead and 2. getting the target from the base output for the extended chunks?

[Q3-3] If it is all about the two aforementioned points, why not simply apply batch expansion? We can have the first half of the batch unmasked and have the look-ahead causal chunked attention masks, while the second half (also copied from the first half) partially masked and having the corresponding causal chunked mask that is equivalent to CADA's design. The batch-expanded version seems more natural to me, since it requires no length expansion for each utterance and avoids specialized convolution operations (split and concat in Figure 2-(b)).

---

> ### Author Response · Authors · 2025-11-20
> **Rebuttal by Authors**
>
> We appreciate reviewer TKjS for detailed comments and questions.  We address all questions below and add the CADA pseudo code in the [post](https://openreview.net/forum?id=ISSxXXiu3w&noteId=gIe41MgToK).  Please let us know if you have more questions or concerns.
>
> **[Q1-1]**
> The modeling unit in the downstream task, i.e., ASR, is referring to phoneme or subword tokens. The number of phonemes or subword tokens are varied from tens to few thousands in an ASR system, which are far less than the high resolution FSQ codebook size ( a few millions) proposed in this study. Table 4 demonstrates phone purity (phn pur.) and phone-normalized mutual information (PNMI). Both metrics are used to measure how a FSQ unit maps to a phoneme. The high values indicate that most frames associated with an FSQ unit belong to the same phoneme (obtained by force alignment), for example the codebook with size 6,834,375 (fifth row in Table 4).
>
> **[Q1-2]**
> `Inspired .. (Liu et al, 2021)`
> In CAAT work, their speech encoder is a chunk encoder with a lookahead chunk. Details could be found in their paper's repo at https://github.com/danliu2/caat/tree/main/.
>
> `naive chunk SSL algorithm`
> Thanks for pointing this out out.
> In this work, we try to propose a SSL algorithm for both streaming and offline speech processing applications. A simple solution in our mind is
> to adopt chunk encoder, which could process partial audio input required streaming application, and predicting information in the last chunk
> based on the information from previous chunks. The idea is simple but not efficient, since we need to calculate the $N$ audio chunks and only get an
> error signal from the last chunk, in which the model is trained to predict the FSQ tokens. So we call this *naive chunk SSL algorithm*. It is also
> new in this work but not efficient. The proposed CADA based chunk SSL training is more efficient than the naive chunk SSL. We will update the paper in the revision to make the *naive Chunk SSL* more clear.
>
> **[Q1-3]**
> In figure 1, *B2* is the consecutive chunk of *B1* and we copy *B2* to the end of utterance as the extended chunk (*E1*) of *B1*. The SSL will predict the masked frames in *E1* based information from *B1*. While *B3* is the last chunk in this utterance, there is no consecutive chunk available or SSL can predict. So there is no extended chunk for *B3*.
>
> **[Q1-4]**
> As mentioned in the reply for [Q1-2], the *naive SSL algorithm* is only the first prototype we have in our mind and it is low efficiency, we didn't build any model based on it.
>
> **[Q1-5]**
> Thanks for pointing out of this. $\mathbf{X}$ is the input sequence to a conformer layer. We will update line 147 to make it more clearly as below
> *Assuming $N$ is the number of input frames $\mathbf{X}$ and is a multiple of chunk size C. The length of the augmented input sequence of $\mathbf{X}$ is N′= 2N−C ...*
>
> **[Q1-6]**
> As explained in [Q1-3], *E1* is extended chunk of *B1* and it is a copy of *B2*. The model is to predict masked frames *E1* based on context information from the previous chunks, i.e. *B1*.

---

> > ### Author Response · Authors · 2025-11-20
> > **Rebuttal By Authors (cont.)**
> >
> > **[Q2-1]**
> > Please refer to replies in [Q1-2] and [Q1-4].
> >
> > **[Q2-2]**
> > It's a great question. The chunk based SSL helps to unify the streaming and offline pre-training, which is new compared with previous work and is one of main contributions in this work. High resolution FSQ is another contribution. FSQ codebook achieves similar results as other codebook, such as BEST-RQ in our experiments (See below).  Our study shows that when we increase the codebook size to a few millions, the SSL would help to boost the downstream task further.  We proposed grouped FSQ to leverage high resolution codebook in the pre-training stage, while it is hard to apply to other
> > quantization approaches, such as  BEST-RQ or k-means. Below is comparison of results with BEST-RQ and FSQ on the Librispeech dataset. We list base configure systems with different initialization. The *Random* means no pre-training is applied. The second and third rows are models initialized with pre-trained models with BEST-RQ and FSQ with moderate codebook sizes, i.e. 8k and 50k respectively. The results show there is no big difference for BEST-RQ and FSQ if the codebook size is moderate and FSQ could outperform all initialization if the codebook size increases to a few millions (6m).  We will include the new results in our revision.
> >
> > | Model                     | dev-clean | dev-other | test-clean | test-other | ave. |
> > |---------------------------|-----------|-----------|------------|------------|------|
> > | Random                    | 2.3       | 5.8       | 2.4        | 5.8        | 4.1  |
> > | Chunk SSL + BEST-RQ (8k)  | 2.1       | 5.3       | 2.2        | 5.3        | 3.7  |
> > | Chunk SSL + FSQ (50k)     | 2.1       | 5.3       | 2.2        | 5.1        | 3.7  |
> > | Chunk SSL + FSQ (6m)      | 2.0       | 5.0       | 2.1        | 4.9        |  3.5 |
> >
> >
> >
> > **[Q2-3]**
> > Thanks for your suggestion, we will include more results from literature into Table 3 in the revision version.
> >
> > **[Q2-4]**
> > Yes, all results in the first part of tables are from literature. As far as we known, no code or recipe is released for the BEST-RQ paper. We acknowledge your concerns about consistency. We try our best to match previous system. For example, the amount of parameters in this work is similar to w2v2 paper. Also, we tried our best to reproduce BEST-RQ system and the results are shown in the reply for [Q2-2].
> >
> > **[Q3-1]**
> > One of the main contribution of this work is we proposed chunk based SSL training. It includes two folds. First is to conduct the pre-training by predicting the missing information in the last chunk based information from all previous chunks (naive chunk SSL). Second, CADA is
> > incorporated to achieve efficient pre-training.
> >
> > **[Q3-2]**
> > In speech processing, we believe it is important to have both left and right context to generate high quality context representation. In chunk based SSL pre-training, we mask partial frames (first half chunk in the right-most chunk) and predict the masked frame indices, which are from FSQ. The prediction is achieved with left context (base chunk and all chunks before base chunk) and limited right context (second half of the right most chunk).  In *copy and append* (CADA), we consider the extended chunks as the right most chunks, for example, *E2* is the right-most chunk for (*B1*, *B2*) chunks in Figure 1.  All masking operations are only applied to the frames in the extended chunks. Hence, we could get a clean left context all the time.
> >
> > **[Q3-3]**
> > Please refer to replies in [Q3-2].

---

> ### Comment · Reviewer_TKjS · 2025-11-28
>
> Thank you for the detailed response and for conducting the additional experiment on the effect of FSQ.
>
> With the clarifications provided, I now have more confidence in the work. My earlier confusion mainly stemmed from the ambiguity of the word “extended,” which led to some misinterpretations. The authors’ explanations satisfactorily address [Q1-3, Q1-6; Q3-2, Q3-3]. On the other hand, I believe the additional experiment still lacks a conventional, self-reproduced SSL baseline, which is necessary for a clearer comparison.
>
> That said, even after the clarification, I still view CADA primarily as a technical training-efficiency improvement. The current version of the paper does not sufficiently demonstrate how significant or practical this efficiency gain is. The contribution would be strengthened if the authors could explore more realistic or competitive scenarios—for example, applying CADA to recent SOTA ASR or ST systems (e.g., Whisper, Parakeet) and showing that it can make their encoders causal in an efficient and scalable manner. Alternatively, the authors could further investigate and highlight interesting behavioral differences between chunked SSL and conventional SSL models.
>
> Overall, I will raise my score to 4. However, I still believe the paper requires major revision, particularly regarding presentation clarity and experimental completeness.
>
> (The review edit button seems to be broken. I will update the score once it comes back.)

---

### Author Response · Authors · 2025-11-20
**Pseudo-code for CADA Augment With Attention and Convolution modules**

```
Algorithm CADA_With_Attn_Conv_GroupMasking(X, C, mask_rate):
    # inputs
    #    X: audio_signal,
    #    C: chunk_size
    #    mask_rate: float, value in [0, 1]
    # --------------------------------------------------------------
    # 1. Compute parameters
    # --------------------------------------------------------------
    M      ← floor(N / C)              # number of base chunks
    N_copy ← N − C                     # length of copied region
    N_aug  ← N + N_copy                # final augmented length

    # --------------------------------------------------------------
    # 2. Create mask ONLY for extended copied chunks
    # --------------------------------------------------------------
    mask ← zeros_vector(N_copy)        # 1 = masked, 0 = keep

    For copied_idx = 1 … M−1:
        local_start ← (copied_idx - 1) * C

        span_len   ← floor(C * mask_rate)
        span_start ← random_integer(local_start,
                                    local_start + C − span_len)

        For t in span_start … span_start + span_len − 1:
            mask[t] ← 1                 # only mask extended copied region

    # --------------------------------------------------------------
    # 3. Build CADA-augmented sequence and zero-mask
    # --------------------------------------------------------------
    BaseChunks   ← [ X[k*C : (k+1)*C] for k = 0 … (M−1) ]
    CopiedChunks ← BaseChunks[1 : M−1]

    X_aug ← concatenate(BaseChunks, CopiedChunks)     # length N_aug

    # group_FSQ_labels = FSQ indices per sub-codebook channel (R channels)
    group_FSQ_labels ← FSQ_encoder(CopiedChunks)

    X ← X_aug

    # Apply mask: zero-out masked frames in copied region
    For t = 0 … N_copy−1:
        If mask[t] == 1:
            X[N + t] ← 0

    # --------------------------------------------------------------
    # 4. Build attention routing mask α by Equation (2)
    # --------------------------------------------------------------
    α ← init_attn_mask(N_aug, N_aug, C)   # based on Equ 2

    # --------------------------------------------------------------
    # 5. L Conformer layers
    # --------------------------------------------------------------
    For layer ℓ in 1 … L:

        # ----- 5a. Q/K/V -----
        Q ← Wq_ℓ * X
        K ← Wk_ℓ * X
        V ← Wv_ℓ * X
        ζ ← 1e6
        # ----- 5b. Masked self-attention -----
        For i = 0 … N_aug−1:
            max_score ← max_j( β·dot(Q[i],K[j]) − ζ·(1−α[i,j]) )
            denom ← 0
            For j = 0 … N_aug−1:
                s[j] ← exp( β·dot(Q[i],K[j]) − ζ·(1−α[i,j]) − max_score )
                denom += s[j]
            For j = 0 … N_aug−1:
                A[i,j] ← s[j] / denom
            X_att[i] ← Σ_j A[i,j]·V[j]

        X ← X_att

        # ----- 5c. Chunk-based convolution -----
        X_new ← zeros_like(X)

        For ci = 0 … M−1:

            s ← ci*C
            e ← s+C
            X_ci ← X[s:e]

            If ci == 0:
                X_lc ← PAD repeated L_lc times
            Else:
                X_lc ← X[s−L_lc : s]

            X_rc ← X[N+s : N+e] + PAD repeated L_rc times

            X_seg ← concatenate(X_lc, X_ci, X_rc)

            X_conv ← Convolution_ℓ(X_seg)

            X_new[s:e]       ← X_conv[0:C]
            X_new[N+s:N+e]   ← X_conv[C:2C]

        X ← X_new

    # --------------------------------------------------------------
    # 6. Group Masked FSQ Loss (only on masked extended frames)
    # --------------------------------------------------------------
    loss ← 0
    count ← 0
    O ← X[N:]   # outputs corresponding to copied/extended region

    For t = 0 … N_copy−1:
        i ← t
        If mask[i] == 1:
            o_i ← O[i]                     # model output for frame i
            µ_i ← group_FSQ_labels[i]      # FSQ indices per sub-codebook channel
            For r = 1 … R:                 # R sub-codebook channels
                sub_idx ← µ_i[r]
                e_r_sub ← embedding_r[sub_idx]
                numerator ← exp(dot(o_i, e_r_sub))
                denom ← 0
                For j = 1 … K_r:
                    denom ← exp(dot(o_i, embedding_r[j])) + denom
                loss ← -log(numerator / denom) + loss
            count += 1

    loss ← loss / count

    return loss
```

---

### Author Response · Authors · 2025-12-03
**Rebuttal Summary**

We thank reviewers valuable feedbacks and suggestions. We appreciate all reviewers agree that this work proposes an efficient way to unify self-supervised pre-training for both streaming and offline speech applications. We want to use this opportunity to emphasize two main contributions in this work again:
- A unified framework, i.e., a chunk based SSL pre-training algorithm, to both streaming and offline speech applications, such as ASR and speech translations. Though chunk based training has been used for fine-tuning by many other works and we introduce chunk based SSL pre-training with the help of the novel CADA algorithm.  The proposed method can achieve very competitive results with one model for both streaming and offline applications. As far as we know, our work is the first to achieve this goal.
- We introduce FSQ for the speech SSL pre-training and demonstrate the high resolution codebook is critical to achieve good performance.  The group based masked prediction loss is introduced to overcome the difficulties of optimizing the high resolution codebook.

We have made following major modifications/updates in the revised paper based on reviewers valuable feedbacks.
1. Add Algorithm 1 (CADA Computation With Attention and Convolution Modules.) in Appendix.
2. Add Table 9 in Appendix to compare BEST-RQ and FSQ with different codebooks.
3. Correction on Equ. 1

---

### Meta-Review · Area_Chair_28Yf · 2026-01-04

**Summary:**

The paper proposes a chunk based SSL framework, approaches to efficiently train it to enable scalable training, including large vocabulary sizes. Overall, reviewers agree that the method is interesting, and achieves good results.

The main issues raised during the review:
1. Correctness: Setting logits to 0 instead of large negative values, causing context leakage.
2. Novelty / missing existing literature.
3. Insufficient analysis of features.
4. Presentation.
5. Fair comparison with existing literature.

While the authors addressed a lot of the concerns that were raised, given the limited novelty and lack of extensive comparisons and analysis, it is unlikely that the rebuttal would have changed the scores enough to result in the paper being accepted.

**Reviewer Concerns:**

1. The authors commented that this was due to a typo and that they use a large scalar to offset softmax.
2. The authors highlight chunk-based pre-training to unify streaming and non-streaming. However, this is an incremental improvement over existing strategies (e.g., BestRQ uses streaming pretraining, focusing on a fully online version; it is unclear whether the model compares favorably with BestRQ at the same latency). The use of high resolution FSQ is another novelty, but the gains in quality here are small.
3. The authors argue that the current analysis still shows that over-representation can help with quality. But the main feedback about purity and analysis remains. Other questions around representation learning are more general in nature (why not Mel), and were answered.
4. The authors addressed some of the specific concerns about presentation that were raised.
5. The authors use resource constraints as a reason for limited comparisons. They also try to explain away some of the differences (compared to offline BestRQ, which performs significantly better); having results to corroborate this would have strengthened the arguments. Per the authors, evaluations on other tasks are to be addressed in future work.

Some questions by Reviewer Ve4f were left unanswered (about infinite chunk size, comparison when using causal self attention, etc.).

**Reviewer Scores:**

Reviewer scQb (Shinji Watanabe): 2 -> 4.

Reviewer TKjS (Liang-Hsuan Tseng): 2 -> 4.

Reviewer Ve4f (Gene-Ping Yang): 4 -> 4.

Reviewer kQdN (Xiaofei Wang): 6 -> 6.

---

### Decision · Program_Chairs · 2026-01-26

Reject